# Insight on Poleward Moisture and Energy Transport into the Arctic from ERA5

**Weifu Sun** [1,†] , **Yu Liang** [2,3,†] , **Haibo Bi** [2,*] , **Yujia Zhao** [4] , **Junmin Meng** [1,5] **and Jie Zhang** [1]

[1]  First Institute of Oceanography, Ministry of Natural Resources, Qingdao 266061, China;
sunweifu@fio.org.cn (W.S.); mengjm@fio.org.cn (J.M.); zhangjie@fio.org.cn (J.Z.)

[2]  Center for Ocean Mega-Science & Key Laboratory of Marine Geology and Environment,
Institute of Oceanology, Chinese Academy of Sciences, Qingdao 266071, China; liangyu17@mails.ucas.ac.cn

[3]  Key Laboratory of Marine Geology and Environment, University of Chinese Academy of Sciences,
Beijing 100049, China

[4]  College of Marine and Information Space, China University of Petroleum (East China),
Qingdao 266580, China; z20160071@s.upc.edu.cn

[5]  Oceanic Telemetry Engineering and Technology Innovation Center, Qingdao 266061, China

\*  Correspondence: bhb@qdio.ac.cn; Tel.: +86-0532-82898935

†  These authors contributed equally to this work.

**Abstract:** With the new-generation reanalysis product (ERA5), the spatiotemporal characteristics of poleward atmospheric moisture and energy transport over the past four decades (1979–2020) were examined. The main channels of atmospheric transport entering the Arctic in the Northern Hemisphere include the Chukchi Sea at 170° W, Baffin Bay at 50° W, North Atlantic at 0° E, and central Siberia at 90° E. Summer (winter) is characterized by high moisture (energy) transport across 70° N. No clear trend in moisture transport was found, whereas the winter and spring energy transport are declining significantly at a rate of $-7.31 \times 10^5$ W/m/a (99% confidence) and $-6.04 \times 10^5$ W/m/a (95% confidence), respectively. Meanwhile, an increasing trend was found in summer ($4.48 \times 10^5$ W/m/a, 95% confidence) and autumn ($3.61 \times 10^5$ W/m/a, not significant). The relationship between atmospheric moisture and energy transport and different large-scale atmospheric circulation patterns, including the Arctic Oscillation (AO), North Atlantic Oscillation (NAO), and Dipole Anomaly (DA), was explored. Among them, DA was identified as the most favorable pattern in relation to moisture and/or energy intrusion into the Arctic. As a result, the surface air temperature increases are more pronounced over most of the central Arctic under the regulation of DA.

**Keywords:** moisture transport; energy transport; arctic

## 1. Introduction

Since the middle of the 20th century, the Arctic has experienced the most intense warming in surface air temperature (SAT), with a warming rate as high as 1.2 °C/10a [1,2]. Known as Arctic amplification (AA) [1,3–5], the Arctic SAT is more than twice the global average warming rate, with a rate of more than 4 times that in winter [3,4]. Along with the significant increase in temperature, the declining trend of Arctic sea ice cover is accelerating [6,7]. Surprisingly, from December 2015 to February 2016, the Arctic surface temperature set the warmest record ever observed, approximately 0.7 °C higher than the previous winter (2011–2012). Then, the warmest record was reset twice consecutively during the following two winters of 2016–2017 and 2017–2018 [7]. Not only can the rapid warming of the Arctic cause the rapid melting of sea ice, but it can also affect the weather and climate in the middle and low latitudes of the Northern Hemisphere by changing the characteristics of atmospheric circulation [8,9].

The causes of the occurrence of AA and the rapid retreat of sea ice are complex. Among them, the atmospheric downward longwave radiation (DLR) anomaly connected

to poleward moisture intrusion plays a dominant role. With ERA-Interim reanalysis data, Gong et al. [10] found that the increase in poleward moisture flux leads to the enhancement of downward longwave radiation (DLR). Their study demonstrated that enhanced DLR serves as the primary contributor to the winter warming trend in areas north of 70° N. In a case study, Chang et al. [11] found that the increased northward moisture transport in the Arctic region on 29 December 2015 resulted in explosive warming through cloud radiative forcing and non-adiabatic action. In addition to reanalysis data, remote sensing observations have also been used to study moisture transport, although the data were available for a short period. For example, Johansson et al. [12] used remote sensing data from Atmospheric Infrared Sounder (AIRS) and the Moderate Resolution Imaging Spectroradiometer (MODIS) and found that moisture transport would cause thicker clouds, which increases the surface temperature in the Arctic by inducing a positive DLR anomaly. In another study, by analyzing the remote sensing record from AIRS, Boisvert et al. [13] affirmed that moisture transport in spring facilitates the earlier melting of Arctic sea ice. In sum, poleward moisture transport is deemed a crucial factor leading to the appearance of AA and thereby significant losses in Arctic sea ice cover.

The atmospheric energy originating in lower latitudes can be transported to the Arctic, contributing to a rise in air temperature over the high latitudes of the Northern Hemisphere [14]. The influence of atmospheric energy transport on Arctic temperature extends from the sea surface to the deep troposphere [14]. Moisture and heat transport from lower latitudes to the Arctic can be influenced by the north–south gradient of humidity and temperature, respectively [15]. In addition, as the transport of moisture and energy is driven by wind, the behavior of the poleward transport of moisture and energy is regulated by large-scale circulations [16], such as the Arctic Oscillation (AO), North Atlantic Oscillation (NAO), and Dipole Anomaly (DA). Synoptic cyclones are a central component in maintaining the global atmospheric energy and moisture budgets. Hence, cyclones moving into the Arctic also play an important role in carrying water vapor and energy [17–19].

Under the background of rapid Arctic warming, a comprehensive understanding of the spatiotemporal characteristics of poleward transport of moisture and energy to the Arctic region is needed. Previous studies analyzed the spatial and temporal distribution characteristics of water vapor transport into the Arctic, but due to differences in datasets, study periods, and calculation methods, different studies led to diverse conclusions [17,19,20]. There is also a lack of studies on the spatiotemporal characteristics of total energy transport despite abundant research on moisture flux. Moreover, the role of the three typical large-scale atmospheric circulation modes (AO, NAO, and DA) in connection with poleward atmospheric transport remains unclear, which is discussed in this research. In the present study, meridional fluxes in terms of atmospheric moisture and energy across 70° N were investigated based on the new-generation reanalysis data, namely, ERA5. This study is organized as follows. Section 2 describes the data and methods. Section 3 presents the spatial distribution, seasonal variation, and long-term trends of poleward transport flux of atmospheric moisture and energy. The effects of large-scale atmospheric circulation modes on the transport of moisture and energy are discussed in Section 4. The last section concludes this study.

## 2. Materials and Methods

### 2.1. Data

#### 2.1.1. Large-Scale Atmospheric Index

The large-scale atmospheric indices, the Arctic Oscillation (AO) and North Atlantic Oscillation (NAO), are provided by the NOAA Climate Prediction Center (CPC), which are available at https://www.cpc.ncep.noaa.gov/products/precip/CWlink/daily_ao_index/teleconnections.shtml (accessed on 29 March 2022). The AO can be characterized as an exchange of atmospheric mass between the Arctic Ocean and the surrounding zonal ring centered at ~45° N [21]. The NAO consists of a north–south dipole of anomalies, with one center located over Greenland and the other center of the opposite sign spanning the

central latitudes of the North Atlantic between 35° N and 40° N. The Dipole Anomaly (DA) index was provided by Bingyi Wu of Fudan University (2021, personal communication). The DA index represents the second-leading mode of the SLP anomaly in the Arctic north of 70° N [22]. All of these indices have a daily temporal resolution and are averaged over months of different seasons.

### 2.1.2. Sea Ice Concentration

We used the satellite-derived daily sea ice concentration (SIC) provided by the National Snow and Ice Data Center (NSIDC) to investigate the relationship between the transport of moisture/energy and SIE variations, which was obtained from https://nsidc.org/data/nsidc-0079/versions/3 (accessed on 29 March 2022) [23]. SIC fields are available on a polar stereographic projection and are derived from the SMMR, the SSM/I, and the SSMIS by applying the bootstrap algorithm [23]. The latest version (version 3.1) of the dataset provides improved consistency between sensors through the use of a suite of daily varying tie points generated from AMSR-E observations. Both datasets have a spatial resolution of 25 km and a temporal resolution of 1 day. Note that sea ice extent (SIE) is defined as the area of the ocean where the SIC is at least 15%.

### 2.1.3. ERA5 Reanalysis Dataset

In this study, ERA5 reanalysis data were utilized to study the spatiotemporal distribution characteristics of moisture transport toward the Arctic. Compared with ERA-Interim, ERA5 represents the fifth-generation reanalysis product of the European Centre for Medium-Range Weather Forecasts (ECMWF). The ERA5 product assimilates more observational data and satellite data and can estimate atmospheric conditions more accurately [24]. Importantly, it can be employed to study the change in moisture transport in the whole Arctic region over a long period with a finer spatial resolution.

ERA5 6 h meridional wind and specific humidity data from 1 January 1979 to 31 December 2020 were used to estimate moisture transport over the Arctic region across 70° N. The datasets are available at https://cds.climate.copernicus.eu/cdsapp#!/dataset/reanalysis-era5-pressure-levels?tab=overview, (accessed on 29 March 2022) [25]. Compared with satellite observation data, the reanalysis data have higher spatial and temporal resolution over a longer time scale and can be used to assess changes in the transport of moisture and energy over the whole Arctic region. The utilized ERA5 datasets include air pressure, wind, humidity, the vertical integral of meridional moisture transport flux, and the vertical integral of total meridional energy transport flux. The spatial resolution of the selected ERA 5 data is 0.25° longitude × 0.25° latitude.

### 2.2. Methods

### 2.2.1. Moisture and Energy Flux into the Arctic

The meridional moisture flux across 70° N affects the local climate of the Arctic [11]. A positive value of meridional moisture flux indicates northward transport, while a negative value is southward. The moisture flux fi between two continuous atmospheric pressure layers $P_i$ and $P_{i+1}$ can be calculated by Equation (1) [26]:

$$f_i = \frac{1}{2g}q_i v_i + q_{i+1}v_{i+1}(p_i - p_{i+1}) \tag{1}$$

where $g$ is gravitational acceleration with a value of 0.98 (m/s$^2$), $q$ is the specific humidity (kg/kg), $v$ is the meridional wind (m/s), $p$ is the atmospheric pressure (hPa), and subscript $i$ denotes a pressure layer in a continuous atmosphere layer.

The meridional moisture flux in the atmospheric layers from the surface to 500 hPa ($F$) was calculated [26].

$$F = \int_{i=1}^{n-1} f_i = \int_{i=1}^{n-1} \frac{1}{2g}(q_i v_i + q_{i+1}v_{i+1})(p_i - p_{i+1}) \tag{2}$$

where $n$ is the number of atmospheric pressure layers. Considering that nearly 98% of atmospheric moisture in the Arctic region is concentrated below 500 hPa of the atmospheric column [27], the vapor flux above 500 hPa is not taken into account. The northward energy transport flux of the whole atmospheric column can be computed according to Equation (3).

$$J_\lambda = \frac{1}{g} \int_0^1 v \left( \frac{1}{2} \boldsymbol{u} \cdot \boldsymbol{u} + c_p T + gz + Lq \right) \frac{\partial p}{\partial \eta} d\eta \tag{3}$$

where $\boldsymbol{u} = (u, v)$ is the wind vector, v and u are north and east wind components, respectively, $c_P$ is the specific heat capacity of wet air at constant pressure, $T$ = temperature, $z$ is potential height, $L$ is condensation specific heat, $q$ is specific humidity, $P$ is atmospheric pressure, and $\eta$ is the vertical mixing coordinate used in the ERA5 atmospheric model.

We used Formulas (1)–(3) to calculate the transport of moisture and energy, and then we compared the estimated results with the corresponding fields in ERA5 and against the existing dataset archived in ERA5. The results show that the estimated results are highly consistent with the corresponding ERA5 fields in both the magnitude and change in all months across various latitudes (e.g., 70° N) during the period 1979 to 2020, which lends credence to the direct use of the moisture and total energy flux field obtained from ERA5. Consequently, we directly utilized the moisture and energy flux data archived in ERA5.

### 2.2.2. Composite Analysis

In this study, composite analysis was used to examine changes in the transport of moisture/energy associated with the different phases of atmospheric circulation conditions. Seasonal anomaly fields (i.e., the differences between the monthly mean values and the corresponding climatology in the period of 1979–2020) were utilized in the composite analysis. Flux fields associated with an index that is one standard deviation greater or smaller than their climatological mean were selected to construct the composite fields.

### 2.2.3. Trend Maps

A trend is represented by the slope of the linear regression, and the significant confidence level of the trend was tested by the Student's *t*-test. If the lower bound of the 95% confidence interval is greater than 0, the trend is significant with a 95% confidence interval. A significant positive trend indicates that the transport is increasing.

### 3. Spatiotemporal Characteristics of Moisture and Total Energy Transport into the Arctic

#### 3.1. Spatial Distribution

In this study, the longitudinal distribution of annual northward moisture and total energy transport of the total atmospheric column across 70° N for 1979–2020 were extracted from ERA5 data. The results are shown in Figure 1. In terms of moisture fluxes, the Eastern Hemisphere is dominated by northward moisture transport (positive values), while the Western Hemisphere has both northward and southward moisture transport, and the transport is more active than that in the Eastern Hemisphere. This distinct characteristic can be attributed to the fact that most of the Eastern Hemisphere is covered by land, while the Western Hemisphere has similar proportions of ocean and land area, and the difference in underlying surfaces (which affects surface–air interactions) leads to greater differences in the direction and intensity of the northward moisture transport. The main channels of moisture entering the Arctic in the Northern Hemisphere include the Chukchi Sea at 170° W, Baffin Bay at 50° W, the North Atlantic at 0° E, and central Siberia at 90° E, which is consistent with the findings of Woods et al. [28]. In addition, Gimeno et al. [29] analyzed the backward trajectory of moisture transport in the Arctic using the Lagrange method based on ERA-Interim reanalysis data. They found that the North Atlantic, North Pacific, and the Labrador Sea were the main channels through which moisture from mid-latitude regions enters the Arctic. Dufour, Zolina and Gulev [17] revealed similar conclusions, although they argued that reanalysis data may overestimate moisture transport.

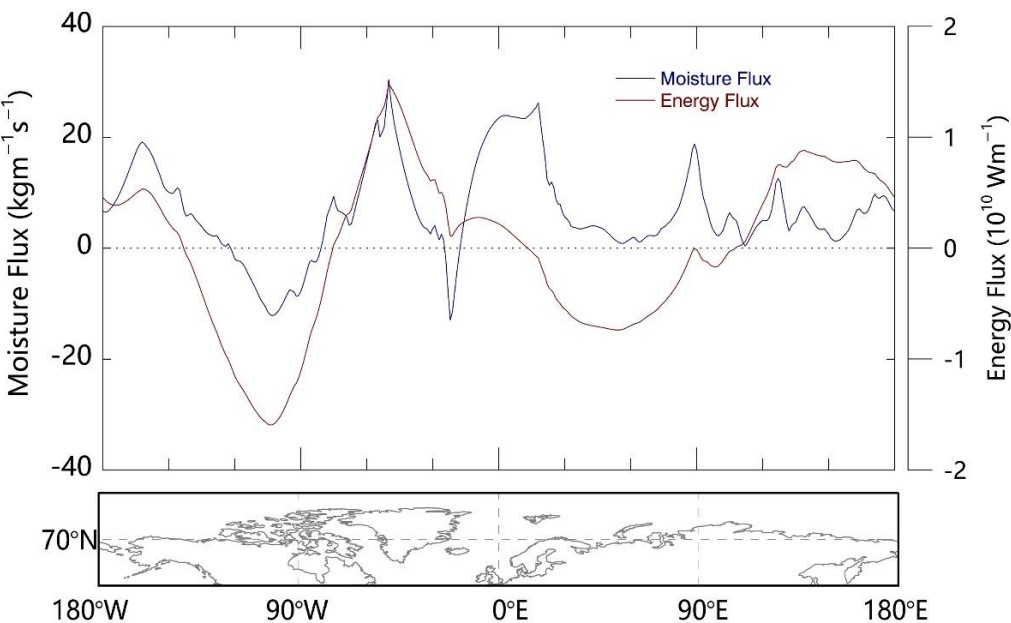

**Figure 1.** Longitudinal distributions of the climatological vertically integrated total meridional energy and moisture fluxes across 70° N during 1979–2020. Note that a positive value of meridional moisture flux indicates northward transport, while a negative value is southward.

The multi-year averaged moisture fluxes in these channels tend to have higher values than 18 kg/m·s, with the highest transport in Baffin Bay (35 kg/m·s). The main channels of southward moisture transport are mainly located in the Canadian Arctic Islands at 100° W and the Denmark Strait at 25° W, where the annual moisture fluxes are above 10 kg/m·s. In addition to the land–sea distribution, moisture transport in the vicinity of these channels can be influenced by other factors. For instance, moisture flux fluxes in the Chukchi Sea, Baffin Bay, and the North Atlantic can exceed 18 kg/m·s and have a wide horizontal distribution, which is mainly affected by cyclones moving northward in the region. Cyclone activity has the ability to carry a large amount of moisture and energy [24].

The southward moisture transport over the Canadian Arctic Islands is more active, which is the result of the joint effect of the local strong north wind and weak cyclone activity [17,30]. The southward moisture transport in the Denmark Strait is caused by the northerly winds associated with the Icelandic low pressure close to the near-surface, where the humidity is highest [18]. The relatively small value and spatial range of moisture transport in central Siberia are mainly due to lower wind speed and humidity in Asia and Europe compared to the ocean. Additionally, using ERA-Interim reanalysis data, Naakka et al. [31] identified that most of the northward moisture transport was offset by southward moisture transport in the Arctic region, and the direction of moisture transport was different as a function of latitude.

Regarding the total energy transport flux, the horizontal distribution of energy transport flux in the Western Hemisphere of the Arctic is largely consistent with the distribution of moisture flux, with a robust correlation coefficient of 0.84 (significant at 99% confidence level, Student's *t*-test). The transport of total energy into the Arctic in the Western Hemisphere is mainly located in the region near the Bering Strait and Baffin Bay. The largest northward transport of energy was detected in Baffin Bay (about $1.5 \times 10^{10}$ W/m), which is mainly related to the northward energy transport caused by the combination of active cyclones [32] and high pressure blocking activity on Greenland Island [33]. The southward energy transport from the Arctic into the lower latitudes mainly occurs in the Canadian Arctic Archipelago, with a larger magnitude ($-1.7 \times 10^{10}$ W/m) and a wider distribution than the northward energy flux. This can be attributed to the heat and kinetic energy fluxes caused by stronger northerly winds. However, the longitudinal distribution of energy fluxes in the Eastern Hemisphere is largely different from that of moisture fluxes and is

less pronounced. The energy entering the Arctic region through the Greenland Sea is relatively small, about $3 \times 10^9$ W/m, while this region is a key channel for the transport of moisture. The lower transport of total energy is related to the offsetting north–south kinetic energy transport in the east–west section of the region and the relatively small amount of energy carried by moisture. In addition, in the region of $30° $ E to $90° $ E, the total energy is transported southward in the opposite direction to that of moisture, which is due to the transport of colder air from Siberia northward into the warmer Arctic Ocean.

Figure 2 shows the spatial distribution of seasonal means of meridional moisture and total energy fluxes into the Arctic obtained from ERA5 averaged over the period 1979–2020. It is shown that the meridional transport of the moisture transport flux has a larger magnitude and more pronounced seasonal variability than the southward component. This contrast may be due to the difference in extra-tropical cyclones at mid-latitudes and in subpolar regions, as well as the drastic changes in humidity during a year. The average seasonal intensity of northward moisture transport in the Arctic decreases in the order: summer (June–August), autumn (September–November), spring (March–May), and winter (December–February) (Figures 2 and 4). Northward moisture transport is significant in summer due to the abundance of moisture, with values even reaching over 50 kg/m·s in the North Atlantic, Bering Strait, and Baffin Bay. These regions are consistent with the main pathways along which moisture enters the Arctic, including the North Atlantic path Greenland, the Scandinavian Peninsula or Norwegian Strait, and the Pacific sector over northwestern Alaska and the Bering Strait [34]. In autumn, most of the moisture flux in the Arctic is less than 20 kg/m·s (Figure 2a). At that time, the moisture fluxes in the Chukchi Sea and Baffin Bay are about 40 kg/m·s, and only the fluxes through the North Atlantic Ocean reach more than 60 kg/m·s (Figure 2a), which is partly caused by the rapid increase in local humidity due to the decline in sea ice [27].

The meridional energy transport is more significant in winter than in other seasons. The main channels of northward energy transport are the East Siberian Sea, Greenland Sea, and Baffin Bay. Additionally, there is a large amount of southward energy flux in the Canadian Arctic Archipelago and the region extending to the North American continent, which is about $3 \times 10^{10}$ W/m. In contrast to the meridional moisture flux, the intensity of northward energy transport in the Arctic increases sequentially in the summer (June-August, $0.62 \times 10^8$ W/m), spring (March-May, $1.00 \times 10^8$ W/m), autumn (September-November, $1.07 \times 10^8$ W/m), and winter (December-February, $1.33 \times 10^8$ W/m) seasons. The meridional energy transport is small throughout the Arctic in summer ($1.04 \times 10^{10}$ W/m), with a maximum value of only one-third that in winter ($2.40 \times 10^{10}$ W/m). In winter, there is stronger northward moisture and energy transport in the North Atlantic Ocean (Figure 2b winter), which may be related to the frequent cyclone activity in this region [35]. In addition, the spatial distribution pattern of the total meridional energy fluxes is essentially the same in winter, spring, and autumn distributions, implying that the intraseasonal variation in energy transport in the Arctic is small.

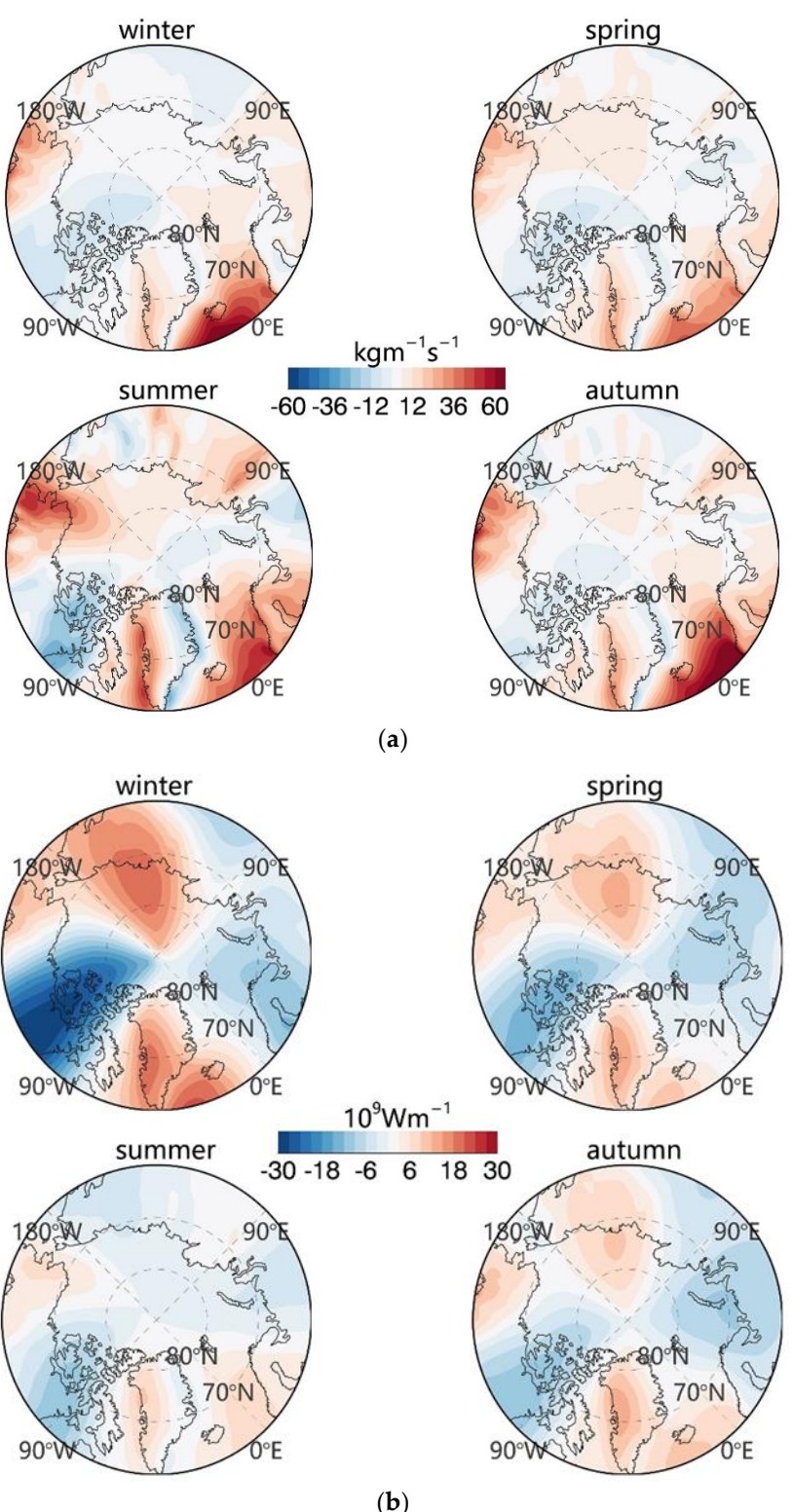

**Figure 2.** Spatial distribution of meridional (**a**) moisture fluxes and (**b**) total energy fluxes across 70° N for the period 1979–2020. The seasons are divided into March–May in spring, June–August in summer, September–November in autumn, and December–February in winter.

### 3.2. Seasonal Variations

The seasonal variations in ERA5 moisture transport and energy transport fluxes across 70° N averaged over the years 1979–2020 are shown in Figure 3. It can be seen that the multi-year average moisture flux and energy flux across 70° N are positive, indicating that the

northward transport of moisture and total energy is stronger than in southward transport. That is, the Arctic as a whole is the convergence region of moisture and total energy. The seasonal variation in moisture transport flux presents distinct intraseasonal variation, with the lowest in winter (about 5 kg/m·s), increasing month by month from January to July, and reaching the highest in July (about 12 kg/m·s). After summer, the moisture fluxes begin to gradually decrease. In general, the maximum value (10.13 kg/m·s) in summer (June–August) is more than twice the minimum value (4.65 kg/m·s) in winter (December, January and February). This may be due to the increased surface temperature in summer, which increases the ability of the atmosphere to retain moisture, the reorganization of the moisture content in the atmosphere, and the humidity of the air compared to winter [35].

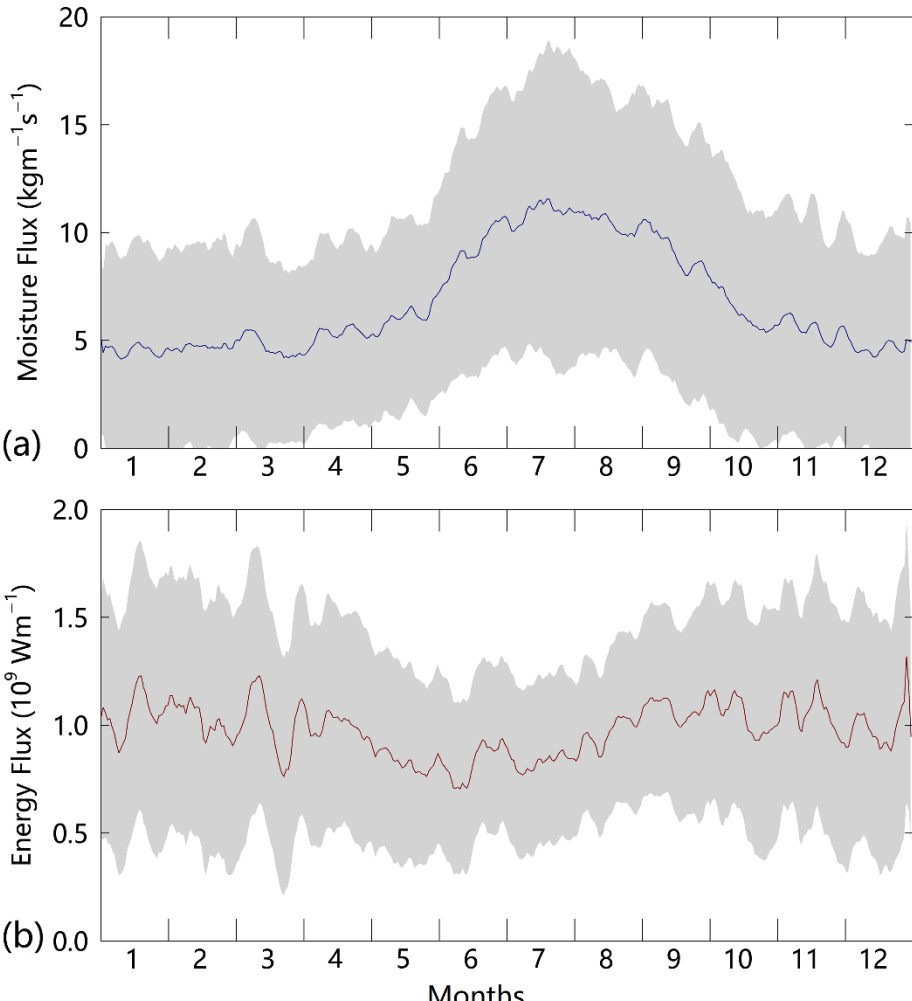

**Figure 3.** Seasonal variations in whole-layer (**a**) meridional moisture fluxes and (**b**) total energy fluxes across 70° N averaged over 1979 to 2020. The fluxes were obtained from the ERA5 product. Shading represents the mean plus/minus one standard deviation.

The daily variation in total energy transport flux is more pronounced and more volatile than that in moisture transport flux. The seasonal variability in the distribution of the total energy transport seems to be opposite to the behavior of moisture transport flux, with a less pronounced inverted unimodal pattern. In the autumn and winter months (October, November, December, January, February and March), the northward energy flux is at a relatively high level, approximately $1.03 \times 10^8$ W/m, and the flux in some months exceeds $1.3 \times 10^8$ W/m. The northward total energy transport is lower in the summer months (May, June, July and August), with an average of approximately $8 \times 10^7$ W/m, with the lowest values occurring in June ($0.58 \times 10^8$ W/m). This distribution pattern

could be a result of increased surface temperatures in the Arctic during summer and a reduced temperature gradient between the polar region and lower latitudes, leading to lower northward energy transport.

### 3.3. Long-Term Trends

Figure 4 plots the time series of total meridional moisture flux and energy flux across 70° N provided by ERA5 from 1979 to 2020. Figure 4a shows that the total meridional moisture flux and the seasonal average across 70° N did not show a significant trend during 1979–2020 but were characterized by large interannual fluctuations, with stronger variability in summer. The trends of moisture flux vary greatly in different seasons. The increasing trend of moisture flux is relatively significant in summer (0.01 kg/m·s/a, not significant), despite the fact that the corresponding moisture flux was decreasing in the channels in the Bering Strait, North Atlantic, and the Denmark Strait, with a magnitude of roughly −1 kg/m·s/a (Figure 5a). Meanwhile, the trend of moisture flux near the Canadian Arctic Archipelago increases in summer (1.1 kg/m·s/a). For the remaining seasons, the trend of moisture flux is small and not statistically significant (Figure 4a). This is consistent with the conclusions of Kapsch, et al. [36] but in contrast to the results of Dufour, Zolina and Gulev [17], who reported that the moisture transport across 70° N showed a decreasing trend from 1979 to 2013.

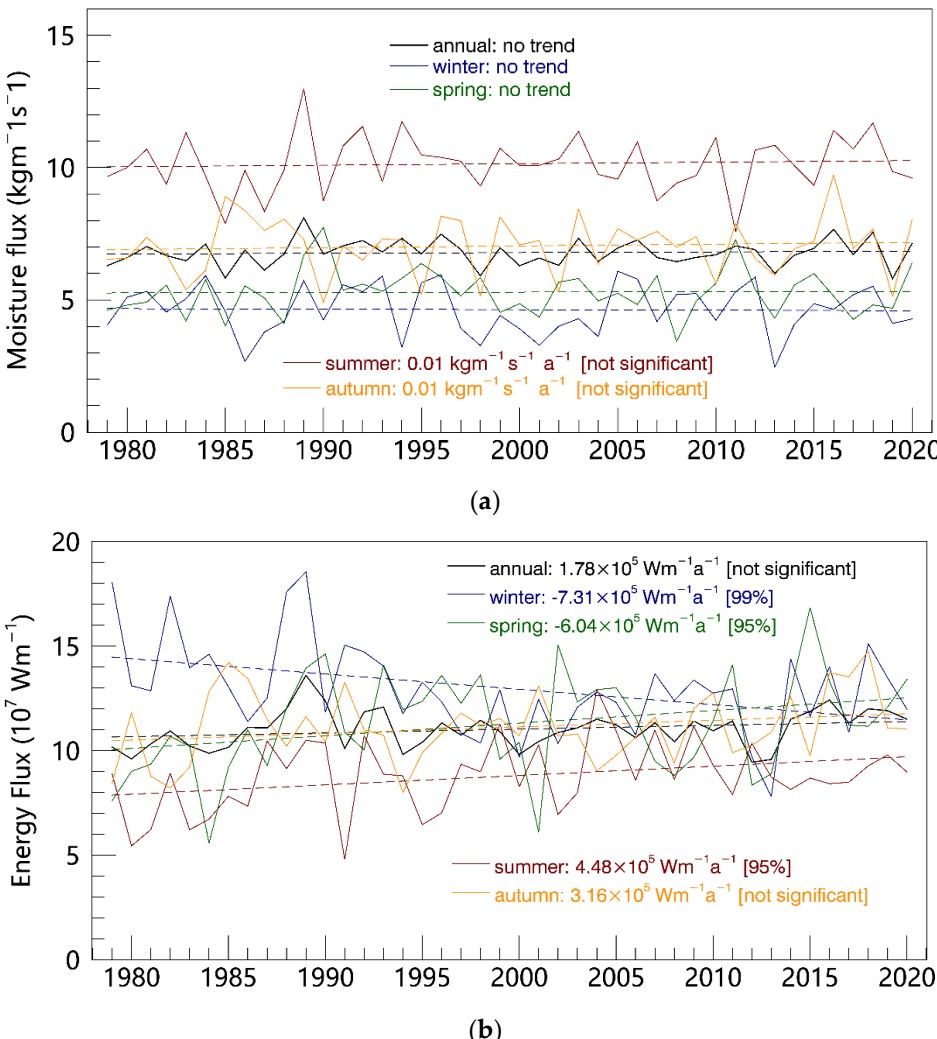

**Figure 4.** The series and trends of (**a**) meridional moisture flux and (**b**) energy flux across 70° N from 1979 to 2020. The record was obtained from the ERA5 dataset.

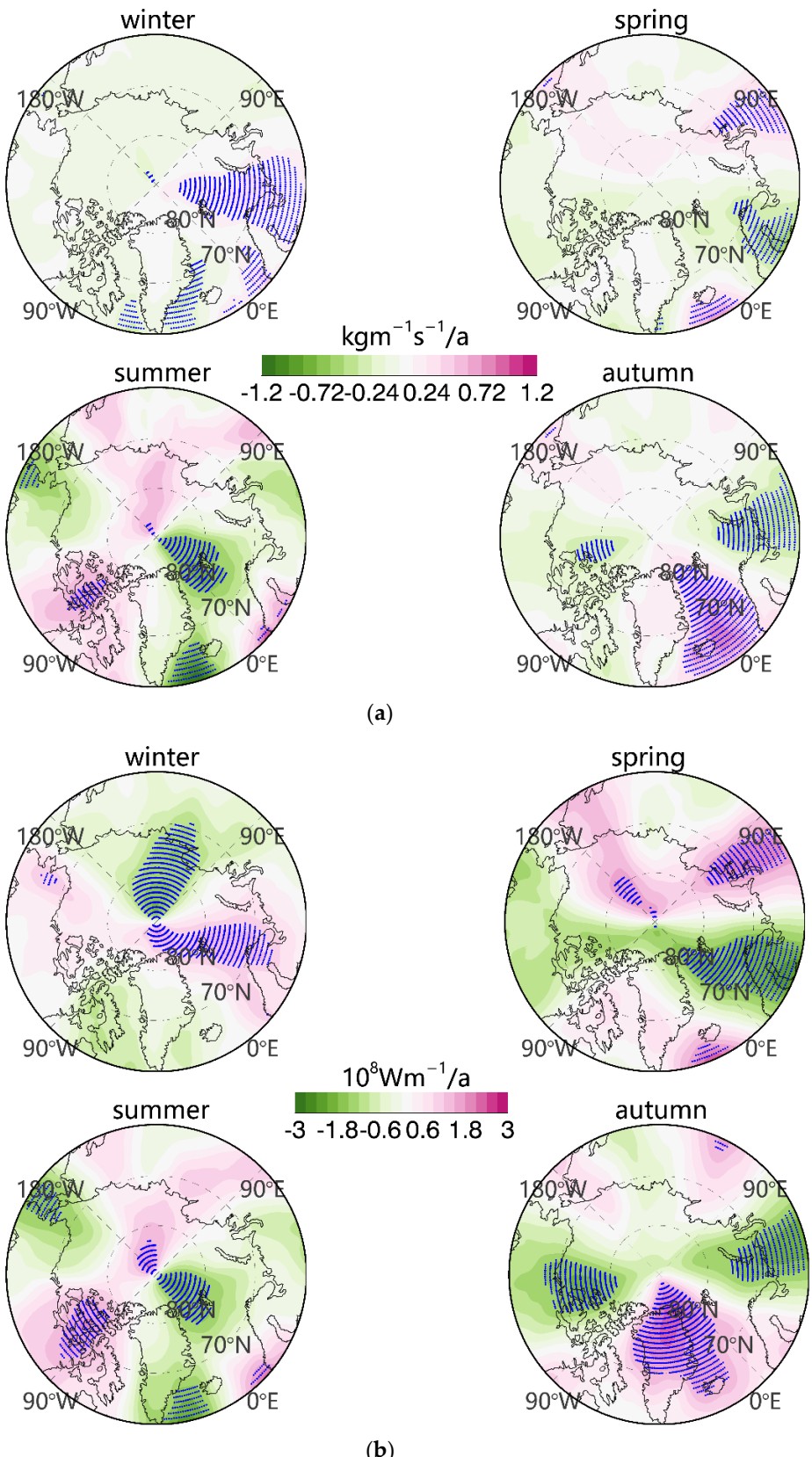

**Figure 5.** Spatial distribution of the trends of (**a**) moisture flux and (**b**) energy flux across 70° N in the ERA5 product over the period 1979–2020. The seasons are divided into spring (March to May), summer (June to August), autumn (September to November), and winter (December to February). The stippled grids denote those with trends significant at the 95% confidence level.

For the total meridional energy transport flux, the annual average meridional energy showed a slight but insignificant positive trend ($1.80 \times 10^5$ W/m/a) from 1979 to 2020, with a significant positive trend in spring and summer, which were $6.04 \times 10^5$ W/m/a and $4.48 \times 10^5$ W/m/a, respectively (Figure 4b). These trends passed the 95% confidence test. In spring, the energy flux increases mainly in the Chukchi and the Barents Sea and decreases significantly in the Nordic Sea (Figure 5b). In summer, the moisture transport flux increases in the vicinity of the Canadian Arctic Islands and decreases in the North Atlantic and the Denmark Strait. By sharp contrast, the mean meridian energy flux in winter significantly decreased from 1979 to 2020 at a rate of $-7.31 \times 10^5$ W/m/a (99% confidence test), with a decreasing trend in the marginal seas of Eurasia (Chukchi Sea, East Siberian Sea and the Laptev Sea), Baffin Bay, and North Atlantic (Figure 5b). By comparing Figure 5a,b, it is found that the spatial distributions of the trends of the total meridional moisture flux and energy flux across 70° N during 1979–2020 are largely similar.

## 4. Spatial Patterns in Poleward Moisture and Energy Transport Associated with Varying Large-Scale Atmospheric Circulation

The total energy and moisture fluxes transported to the Arctic are controlled by large-scale atmospheric circulations [16,37]. In order to explore the relationship between moisture flux and large-scale atmospheric circulation, the Arctic Oscillation (AO), North Atlantic Oscillation (NAO), and Dipole Anomaly (DA) were considered in this study. The averages of the atmospheric circulation modes in winter (December, January and February) were selected in years with higher and lower index values for the composite analysis. The composite results are shown in Figure 6; note that the difference is defined by "higher" minus "lower".

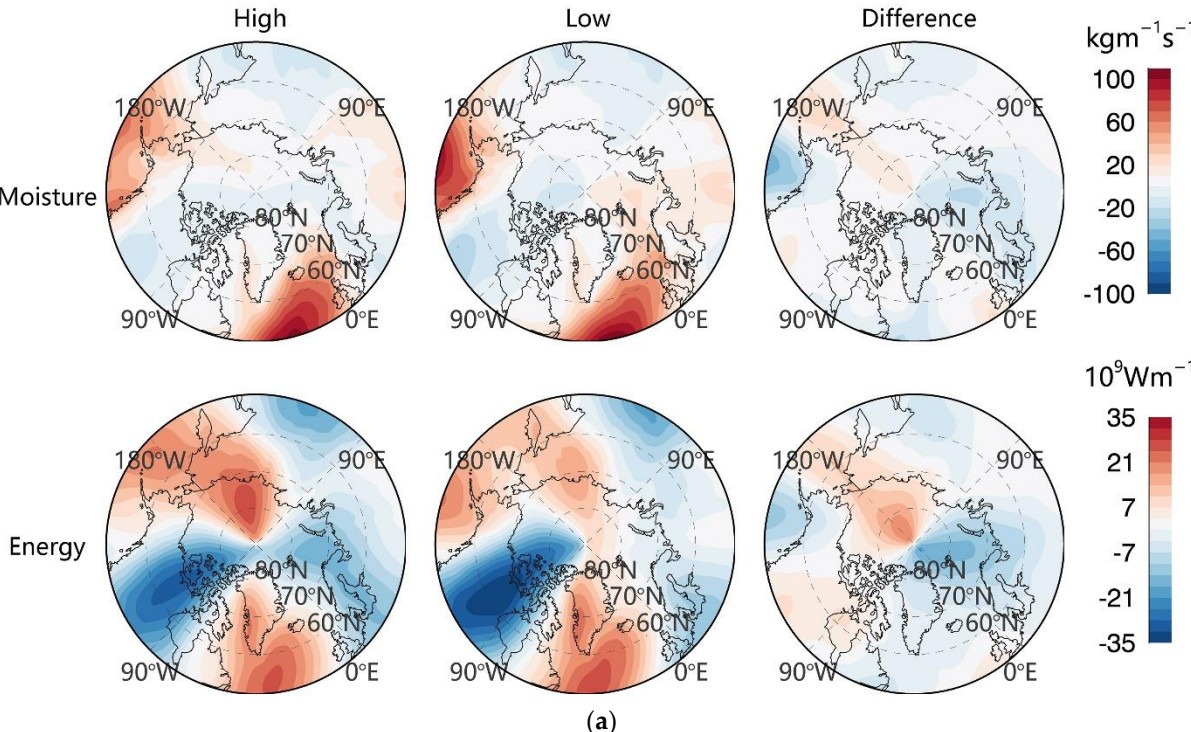

(a)

**Figure 6.** *Cont.*

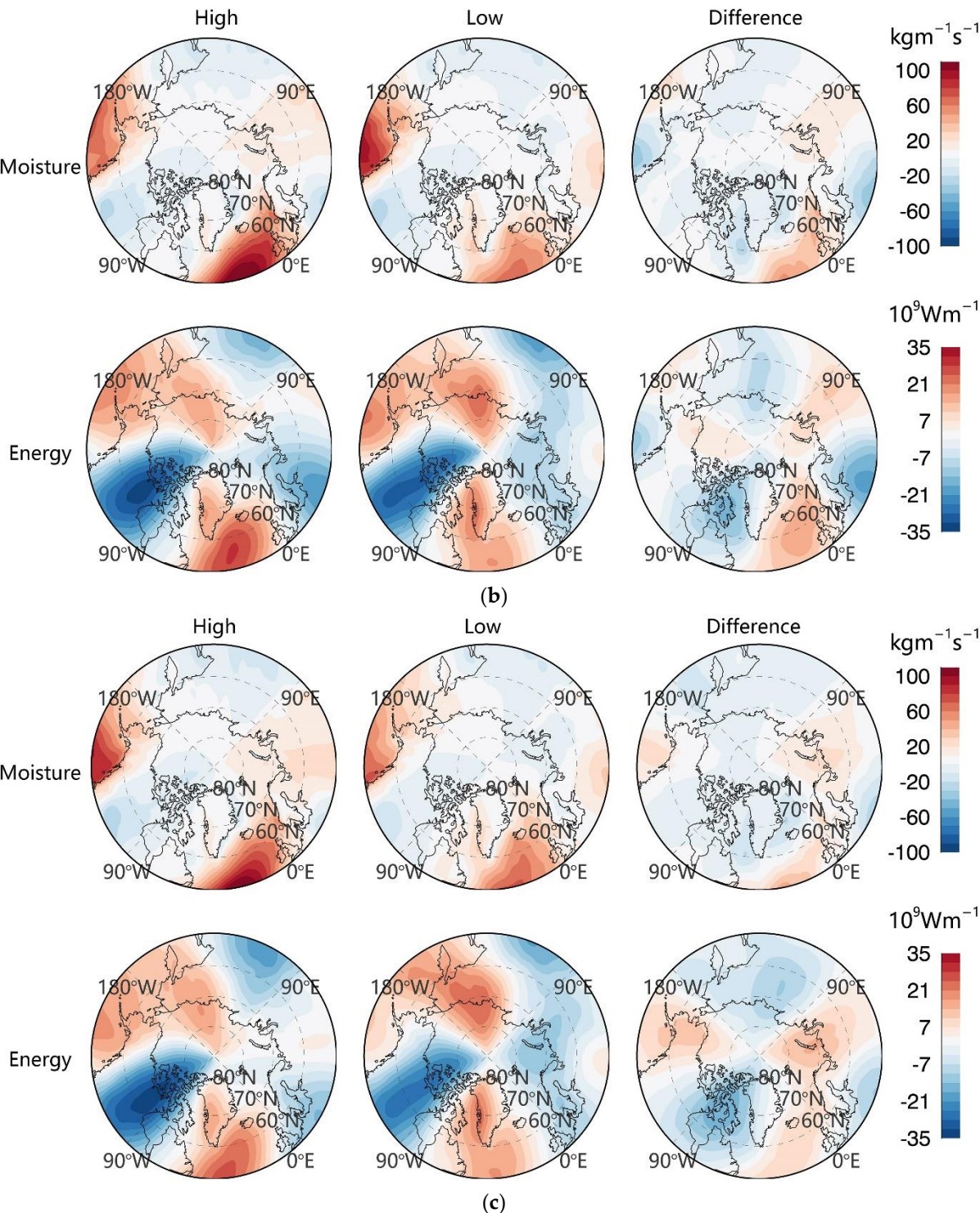

**Figure 6.** Composite analysis of meridional moisture flux and energy flux from ERA5 over the Arctic region based on the most positive and negative large-scale atmospheric circulation indices and their differences: (**a**) AO, (**b**) NAO, and (**c**) DA.

As shown in Figure 6a, the anomalously high AO index (1979, 1982, 1989, 1993, 1996, 2004 and 2019) in comparison with the low AO index (1990, 1991, 2000, 2005, 2006, 2012, 2014 and 2018) shows increased moisture entering the Arctic via the Barents and Kara Seas, as well as increased energy entering the Arctic from the North Atlantic and southward from the Canadian Arctic Archipelago (Figure 6a). The composite map based on the NAO index is largely consistent with the AO composite graph (Figure 6b). When the DA index

was high (1989, 1995, 2000, 2012, 2015, 2016, 2018 and 2020), compared with when the DA index was low (1979, 1985, 1996, 2010 and 2011), the moisture from the Bering Strait to the Arctic increased (Figure 6c). This is related to the strong meridional wind across the pole from the Pacific sector toward the Atlantic sector during the positive phase of DA. Conversely, the moisture flux during the negative phase of DA originates from the North Atlantic and is transported much farther northward in the Arctic. The remarkable difference in energy transport between the positive and negative phases of DA occurs in the Bering Strait.Moisture transport appears to be favorable for a warmer Arctic. Figure 7 indicates that the DA-associated atmospheric circulation can benefit more from a rise in surface air temperature in the Arctic, with an especially evident regime over the Pacific sector (PAS). There has been an approximate rise of 1.9 °C in SAT over the PAS in relation to the enhancement of DA over the past four decades (Figure 7b). Interestingly, the PAS is an area that has experienced significant sea ice loss during the summer months over the past four decades (not shown). The significant NAO-associated SAT increases are mainly concentrated on the surrounding land area over northern Eurasia, the Canadian Arctic Archipelago, and western Greenland (Figure 7c). By comparison, the AO-associated SAT increase is not significant over most of the Arctic (Figure 7d). Combined with the results presented in Figures 6 and 7, the greater effects of DA on Arctic warming are rather apparent compared to either NAO or AO.

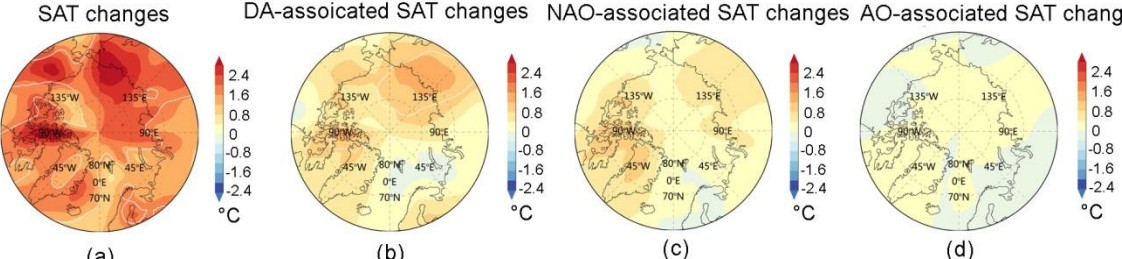

**Figure 7.** SAT trends (**a**) and the trends associated with (**b**) DA, (**c**) NAO, and (**d**) AO. The associated SAT trends are estimated as the regressed slope of the SAT (obtained from ERA5) on different atmospheric indices over the period 1979–2020.

## 5. Influence of Transport of Moisture and Energy on Sea Ice

The transport of atmospheric water vapor and energy to the Arctic will play an important role in regulating the change in sea ice. Olonscheck et al. [38] demonstrated that internal variability in sea ice is primarily caused directly by atmospheric temperature fluctuations. The other drivers together explain only 25% of sea ice variability. Moisture and energy transport bring heat and water vapor to the Arctic. In addition, water vapor transport enhances the local Arctic greenhouse effect, both by the water vapor itself and through the increased cloudiness caused by the transport, leading to positive anomalies in temperature and humidity [39]. Kapsch, Graversen and Tjernström [36] also found that years with a low September sea ice extent (SIE) are characterized by more moisture intrusion into the region where significant ice retreat occurred during spring. This is accompanied by enhanced net longwave radiation plus turbulent fluxes, which is associated with increased humidity and cloud cover.

We analyze the impact of moisture and energy influxes on Arctic sea ice. It is estimated that a correlation exists between the spring SIE and the vertically integrated northward total energy flux (moisture flux) at 60° N with R = −0.43 (significant at the 99% confidence level). The strong correlation indicates that more transport of energy from lower latitudes has a negative effect on sea ice, which is consistent with previous studies. Figure 8 depicts the difference map of SIC between high (one standard deviation greater than the climatological mean) energy influx years and low (one standard deviation smaller than the climatological mean) energy influx years. The difference map is a result of subtracting the averaged SIC during low-transport years from that during high-transport years. Compared with springs

with lower northward energy transport across 70° N into the Arctic, the years with higher energy influx have more concentrated ice in the marginal seas, especially in the Barents Sea, the Greenland Sea, and the Bering Strait.

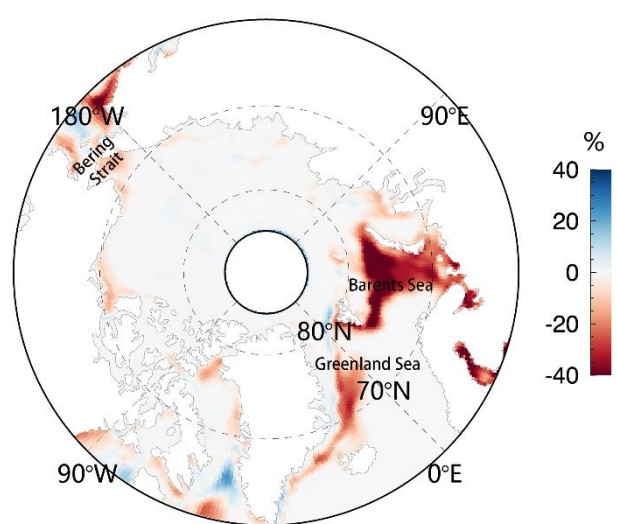

**Figure 8.** Difference map of composite analysis of SIC over the Arctic region based on meridional energy flux across 70° N from ERA5.

## 6. Conclusions

It was found that the poleward energy transport of the Northern Hemisphere atmosphere can advect a large amount of warm and wet air to the Arctic region, which leads to the accumulation of moisture in the Arctic atmosphere and an increase in cloud cover [39]. As a result, the downward longwave radiation of the atmosphere and the greenhouse effect are enhanced, leading to amplified surface warming [36,39]. This study investigated the long-term changes in moisture and energy transport across 70° N over the past four decades (1979–2020) by using the new-generation reanalysis product, namely, ERA5. A preliminary analysis of the influence of energy transport on Arctic sea ice was also conducted.

The results suggest a large interannual variability in moisture transport in different seasons, but no significant trend was found over the investigated period. By contrast, the trend is significant for energy transport, with a rate of $-7.31 \times 10^5$ W/m/a (99% confidence) in winter and $-6.04 \times 10^5$ W/m/a (95% confidence) in spring. At the same time, an increasing trend is encountered in summer ($4.48 \times 10^5$ W/m/a, 95% confidence) and autumn ($3.61 \times 10^5$ W/m/a, not significant). Seasonally, the winter (summer) months are generally characterized by high energy (moisture) transport toward the Arctic. Among the large-scale atmospheric patterns, DA is the most favorable mode leading to a rise in Arctic surface temperature, which is particularly prominent on the PAS side, where large sea ice depletion has been observed from satellite measurements. In addition, enhanced energy transport into the Arctic in spring could favor sea ice cover loss.

An interesting point lies in the significant enhancement of summer energy transport over the past four decades. A partial explanation is the increasing trend in DA, which implies stronger anticyclonic (cyclonic) circulation over the western (eastern) Arctic Ocean. Indeed, the summer atmospheric circulation is primarily associated with a stronger Beaufort high (BH). BH was much more pronounced over the period 2007–2012, when the summer/autumn sea ice loss over the PAS was dramatic. Therefore, an upcoming study will be implemented by our research team concerning the effects of the BH-associated anticyclonic circulation on sea ice decrease in the PAS through the modulation of poleward moisture and/or energy transport.

**Author Contributions:** The first authors, W.S. and Y.L., analyzed the spatiotemporal characteristics of poleward atmospheric moisture and energy transport over the past four decades and prepared the manuscript. The third and fourth authors, H.B. and Y.Z., helped to conceive and design the analysis. The last two authors, J.M. and J.Z. contributed to ERA5 data collection and data processing. All authors have read and agreed to the published version of the manuscript.

**Funding:** The research was funded by the National Key Research and Development Program of China (No. 2018YFC1407202, 2018YFC1407206, 2016YFA0600102), the National Natural Science Foundation of China (41876206).

**Acknowledgments:** We thank ECMWF for providing the ERA5 data.

**Conflicts of Interest:** The authors declare no conflict of interest.

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
