# Peer review of "Insight on Poleward Moisture and Energy Transport into the Arctic from ERA5"

_atmosphere, doi:10.3390/atmos13040616_

Round 1
Reviewer 1 Report
The revised article considers the influx of energy and moisture into the Arctic through 70 N according to the EPA5 reanalysis for 1979-2020. The main part of the article is devoted to the analysis of space-time distributions of integral transport. Spatial distributions, seasonal variations, and a long-term trend of poleward transport of atmospheric moisture and energy are presented. The results show a significant interannual variability of transport and the absence of significant positive trend in integral transport, except for summer and autumn. At the same time, in the spatial distribution of transprt, positive trends were established for energy transport in winter and autumn through the Atlantic part (Fig. 5).
The study of the influence of large-scale atmospheric circulation regimes (AO, NAO and DA) on transports in this paper (Section 4) represents the original part of the study. It has been established that the influence of the DA mode is most noticeable, during which an increase in the surface temperature in the Arctic is noted. Composite transport maps with significant positive and negative deviations of the circulation indices for the
considered modes indicate the pre-Atlantic and Pacific regions of prevailing transport.
In conclusion, we note that this part of the study is of interest and has the potential for further research. This allows to recommend an article for publication.
Reviewer 2 Report
Thank you for addressing all the comments. I do not have any further comments. It is an interesting work. I recommend this paper for publication.
This manuscript is a resubmission of an earlier submission. The following is a list of the peer review reports and author responses from that submission.
Round 1
Reviewer 1 Report
//
The article considers the influx of energy and moisture into the Arctic through 70 N according to the EPA5 reanalysis for 1979-2020. Although a formula is given for calculating transports for individual layers of the troposphere, vertical transport profiles are not considered, and the study is limited to using integral transports in the entire troposphere. The main part of the article is devoted to the analysis of space-time distributions of integral transport. Spatial distributions, seasonal variations, and a long-term trend of poleward transport of atmospheric moisture and energy are presented. The results show a significant interannual variability of transfers and the absence of significant positive trends in integral transfers, except for summer and autumn. This result contradicts previous studies that increased water vapor influx and subsequent growth of DLR (Cao, Y.; Liang, S.; Chen, X.; He, T.; Wang, D.; Cheng, X. Enhanced wintertime greenhouse effect reinforcing Arctic amplification and initial sea-ice melting, Sci Rep. 2017.7, 84624; Gong et al., 2016; Chang et al., 2016) serve as the main factor of winter warming in the northern regions from 70°N.
Such a discrepancy is due to the non-representativeness of the integral transfer for assessing the impact on warming. The reason is that transfers to the Arctic in winter are concentrated in the lower troposphere up to a height of 750 hPa through its near-Atlantic part. The direction of transport in higher layers is from the Arctic, which corresponds to the polar cell of the meridional circulation. In summer, the directions are reversed, with the exception of the Pacific part of the Arctic. (Alekseev et al., 2019). This feature of the spatial distribution of transfers is manifested in the spatial distribution of transfers in Fig. 2 and in the positive trends of energy transfers in winter and autumn through the Atlantic part in Fig. 5 peer-reviewed work.
The study of the influence of large-scale atmospheric circulation regimes (AO, NAO, and DA) on transports in this work (Section 4) takes up less (four times) space compared to integral transports (Section 3). It has been established that the influence of the DA mode is most noticeable, during which there is an increase in surface temperature in the Arctic. Composite transport maps with significant positive and negative deviations of the circulation indices for the modes under consideration practically do not differ from each other and indicate the pre-Atlantic and near-Pacific regions of prevailing transports.
In conclusion of paper, the results of previous studies by other authors are again mentioned and their own contradictory results about negative trends in moisture and energy transfers to the Arctic are presented. The increasing role of the DA circulation mode and the connection of the summer atmospheric circulation with the Beaufort High (BH) are noted, which is supposed to be investigated in the future.
In conclusion, we note that the totality of the considered results does not contain sufficient novelty and reliability, which does not allow us to recommend the work for publication.
Reviewer 2 Report
Review on Atmosphere - 1609141
Manuscript of “An insight of the poleward moisture and energy transport into the Arctic from ERA5”
Authors: Weifu Sun, Yu Liang, Haibo Bi, Yujia Zhao, Junmin Meng, and Jie Zhang.
General comments:
This paper investigates the meridional moisture and energy transport trends in the Arctic, and how this is associated with large-scale circulation such as Arctic Oscillation (AO), North Atlantic Oscillation (NAO), and Dipole Anomaly (DA). Authors found that moisture transport has no distinctive trend while energy transport has decreasing trends, especially in winter and spring, and increasing trends in summer and autumn. They also reported that DA is the most relevant pattern to the moisture and energy transport to the Arctic and increasing surface air temperature is contributing to this.
This paper deals with interesting topics and most of the parts read well. I have a few comments to clarify the points.
Minor comments.
- What is the exact role of DA? In line 364-368, the different phase of DA plays a different role in energy transport in Bering Strait. It is unclear in what phase of DA contributes to sea ice loss. Do you mean that their impacts on sea ice are dependent on the regions?
- It is still not clear what threshold was used to select the high and low AO, NAO, DA. Does high (low) AO refer to the positive (negative) phase? Please include the threshold you used.
- Line 149-154: So, did you use moisture and energy flux data from ERA5 data after comparison? It is ambiguously stated.
- Line 321-322: What do you mean by “upward” trend? Do you mean by “poleward” trend or “positive” trend? It is confusing. Please modify this with a more appropriate word.
- Line 411-413 and Figure 8:
- How did you compute the difference (%)? You said,” higher energy influx has more concentrated ice in the marginal sea….”. But these areas are marked with “red”, which refers to “negative” values, according to your Fig. 8. Does this mean that a higher energy influx is less likely to cause sea ice loss? If so, this sounds contradicting to what you said. Please clarify this.
- And can you mark where the Barents Sea, the Greenland Sea, and Bering Strait are on the map? This way, readers may easily focus on the contents of the text.
- In the data section (section 2.1 and subsections), please include the exact data resources (e.g., URL) and relevant references.
Reviewer 3 Report
I have read over the authors' revisions and believe they have made adequate effort to revise the paper. I would recommend acceptance in its present form.